# Taxane-Induced Neuropathy and Its Ocular Effects—A Longitudinal Follow-up Study in Breast Cancer Patients

**DOI:** 10.3390/cancers15092444

**Published:** 2023-04-25

**Authors:** Nadine Stache, Sebastian Bohn, Karsten Sperlich, Christian George, Karsten Winter, Friederike Schaub, Ha-Vy Do, Martin Röhlig, Klaus-Martin Reichert, Stephan Allgeier, Oliver Stachs, Angrit Stachs, Katharina A. Sterenczak

**Affiliations:** 1Department of Ophthalmology, Rostock University Medical Center, 18057 Rostock, Germany; nadine.stache@uni-rostock.de (N.S.); sebastian.bohn@uni-rostock.de (S.B.); karsten.sperlich@uni-rostock.de (K.S.); friederike.schaub@med.uni-rostock.de (F.S.); ha-vy.do@med.uni-rostock.de (H.-V.D.); oliver.stachs@uni-rostock.de (O.S.); 2Department of Obstetrics and Gynecology, Rostock University Medical Center, 18059 Rostock, Germany; christian.george@kliniksued-rostock.de (C.G.); angrit.stachs@icloud.com (A.S.); 3Department Life, Light & Matter, University Rostock, 18059 Rostock, Germany; 4Institute of Anatomy, Medical Faculty, University of Leipzig, 04103 Leipzig, Germany; kwinter@rz.uni-leipzig.de; 5Institute for Visual and Analytic Computing, University of Rostock, 18059 Rostock, Germany; martin.roehlig@uni-rostock.de; 6Institute for Automation and Applied Informatics, Karlsruhe Institute of Technology, 76344 Eggenstein-Leopoldshafen, Germany; klaus.reichert@kit.edu (K.-M.R.); stephan.allgeier@kit.edu (S.A.)

**Keywords:** breast cancer therapy, taxanes, neurotoxic events, polyneuropathy, retinal layers, corneal nerves

## Abstract

**Simple Summary:**

The study’s purpose was to determine whether neurotoxic signs in breast cancer patients receiving taxane (paclitaxel) chemotherapy correlate with retinal or corneal nerve changes in a longitudinal study combining oncological examinations with advanced biophotonic imaging techniques. Recruited breast cancer patients underwent regular monitoring sessions including the clinical assessment of their quality of life and neurological scores, ophthalmological status, retinal optical coherence tomography, and large area confocal laser scanning microscopy of their corneal subbasal nerve plexus (SNP). The study revealed two key breakthroughs: an observable increase in retinal thickness and the examination of identical corneal SNP large-area mosaics over time. Consequently, advanced biophotonic imaging techniques could represent a powerful diagnostic tool for the objective assessment of the severity of adverse events, wherein oncologists and ophthalmologists could benefit from each other, leading to a better outcome for the patient.

**Abstract:**

A common severe neurotoxic side effect of breast cancer (BC) therapy is chemotherapy-induced peripheral neuropathy (CIPN) and intervention is highly needed for the detection, prevention, and treatment of CIPN at an early stage. As the eye is susceptible to neurotoxic stimuli, the present study aims to determine whether CIPN signs in paclitaxel-treated BC patients correlate with ocular changes by applying advanced non-invasive biophotonic in vivo imaging. Patients (n = 14, 10 controls) underwent monitoring sessions after diagnosis, during, and after therapy (T0-T3). Monitoring sessions included general anamnesis, assessment of their quality of life, neurological scores, ophthalmological status, macular optical coherence tomography (OCT), and imaging of their subbasal nerve plexus (SNP) by large-area confocal laser-scanning microscopy (CLSM). At T0, no significant differences were detected between patients and controls. During treatment, patients’ scores significantly changed while the greatest differences were found between T0 and T3. None of the patients developed severe CIPN but retinal thickenings could be detected. CLSM revealed large SNP mosaics with identical areas while corneal nerves remained stable. The study represents the first longitudinal study combining oncological examinations with advanced biophotonic imaging techniques, demonstrating a powerful tool for the objective assessment of the severity of neurotoxic events with ocular structures acting as potential biomarkers.

## 1. Introduction

Paclitaxel is among the most commonly used cytostatic drugs in breast cancer treatment. In combination with anthracyclines, paclitaxel significantly reduces breast cancer mortality [1]. Chemotherapy-induced peripheral polyneuropathy (CIPN) is one of the most common dose-limiting side effects of taxanes and can significantly impair quality of life (QoL) and may lead to less effective anticancer treatment [2,3,4]. A meta-analysis of 31 studies with 4179 patients showed a prevalence of CIPN of 68.1% measured 3 months after chemotherapy [5]. CIPN can affect the peripheral nervous system from the nerve cell to the axon and all types of nerve fibers. However, paclitaxel mainly causes sensory axonal damage, characterized by numbness, tingling, abnormal temperature sensations, or a burning sensation in the fingers and toes [6,7]. Currently, there is no approved therapy for CIPN and there is a lack of pharmacological prevention strategies. In practice, local cryotherapy (cooling gloves and socks) or local compression are the most common physical methods of prevention, yet the efficacy is conflicting [8]. Detection is based on subjective criteria, which indicate CIPN only when symptoms are present [9]. Therefore, early detection methods are necessary to allow for therapy adjustment to prevent permanent damage. However, current methods of assessing small fibers are invasive, time-consuming, or subjective. 

The cornea is one of the most densely innervated tissues, and corneal sensory nerves are considered part of the peripheral nervous system [10]. Thus, the cornea could represent a diagnostic window for neuropathic changes [11,12,13]. The sensory innervation of the cornea occurs mainly via the trigeminal ganglion in the form of Aδ and C fibers responsible for nociceptive, chemical, and temperature signaling. Although the innervation of the cornea is similar to the receptors in the skin of the distal extremities of the limb [14,15], limited research on corneal nerve fiber changes associated with CIPN exists. Recent studies have shown that confocal microscopy can visualize small fibers of the cornea in vivo and detect peripheral neuropathy [16]. Corneal confocal laser scanning microscopy (CLSM) is an established method for detecting and monitoring diabetic neuropathy [17]. However, there is low evidence for corneal nerve changes associated with cancer therapy and CIPN. In a recent study of 21 patients with upper gastrointestinal cancer, corneal confocal microscopy could detect a small fiber neuropathy which was related to the severity of CIPN [18]. Moreover, Chiang et al. [19] reported reduced corneal nerve fiber and inferior whorl length in paclitaxel-treated patients compared to healthy controls. Riva et al. [20] tested if the combination of corneal nerve parameters assessed by in vivo confocal microscopy (IVCM) is useful to monitor the neurotoxic effects of chemotherapy compared to epidermal nerve quantification. For this purpose, 95 adults with different cancer types were recruited, and neurological examinations, including a total neuropathy score and *IVCM,* were performed before and after chemotherapy [20]. The experimental data confirmed that in vivo confocal microscopy is a helpful, non-invasive tool with promise for the diagnosis of CIPN [20]. However, at the present time, despite a case report [21], no prospective longitudinal studies which monitor breast cancer patients before, during, and after the course of chemotherapy exist for detecting corneal nerves by using advanced biophotonic techniques. In addition, currently, representative selections of CLSM images and valid image analyses are a challenge. 

Apart from corneal sensory nerves, the posterior segments of the eye, especially the retina and the choroid, also represent important targets for the toxicity of anticancer drugs [22]. Anatomically and developmentally, the retina is an extension of the central nervous system and is composed of layers of specialized interconnected neurons [23]. The retina is among the most metabolically active tissues in the body, making it a prime target for the side effects of chemotherapeutic agents, e.g., macula edema, photopsia, and glaucoma [22,24,25]. In many neurological disorders, ocular manifestations precede neurological symptoms, and thus, retinal examination offers a unique opportunity for the early diagnosis of the side effects of neurotoxic drugs [22]. In this context, optical coherence tomography (OCT) provides excellent non-invasive access to the retina.

In order to help guide the appropriate treatment plan for each patient and to have a significant influence on their outcome and QoL, there is a clinical need for the early detection of neurotoxic side effects. To address this, the present study aims at determining whether neurotoxic signs in breast cancer patients receiving paclitaxel (taxane) correlate with retinal or corneal nerve changes. To this end, non-invasive in vivo biophotonic imaging methods such as corneal CLSM and OCT will be applied before, during, and after the cancer therapy regimen. 

## 2. Materials and Methods

### 2.1. Study Design and Flow Chart

The prospective longitudinal study was approved by the Ethics Committee of the University of Rostock in accordance with applicable laws, rules, and regulations on 19 February 2020 (A 2018-0162). All methods were conducted in accordance with relevant guidelines and regulations. All participants underwent informed consent and signed a consent form prior to the conduction of the study in accordance with the Declaration of Helsinki. All patients underwent monitoring sessions after diagnosis and before (T0), during (T1 and T2), and after cessation (T3) of the paclitaxel therapy regimen. The monitoring sessions included the assessment of QoL and neurological scores, ophthalmological status, and anterior and posterior segment imaging (Figure 1). Fourteen patients with newly diagnosed breast cancer and ten healthy controls were recruited. Of the 14 recruited patients, 2 were excluded due to health reasons, therefore, 12 patients and 10 control subjects were assessed. The 12 patients were able to participate in the clinical examinations depending on their therapy. Because of different therapy durations and individual health conditions, not all patients could be examined at all ophthalmological monitoring sessions. Figure 1 summarizes the total number of patients and controls who underwent the monitoring sessions from T0 to T3.

### 2.2. Patient Selection

Patients were eligible if the following criteria were fulfilled: planned paclitaxel-based chemotherapy for ≥12 weeks, aged 18+ years, and willing to complete all study activities. Patients were excluded if <1/3 of planned neurotoxic chemotherapy was applied, or if they had a pre-existing neuropathy due to other reasons (e.g., diabetes mellitus or alcohol consumption). Healthy controls had no history of cancer, no self-reported signs of peripheral neuropathy, and no ocular diseases. 

### 2.3. Assessment of Clinical Characteristics 

Patients’ and controls’ age, BMI, history of the disease (ophthalmological, neurological, gynecological, oncological, and diabetes mellitus), and additional clinical data (e.g., allergies, current medication, and nicotine, drug, and alcohol consumption) were obtained by interview or chart review. Additionally, patients’ tumor-specific data, such as the initial stage, grading and hormone receptor status, and cumulative dose of chemotherapeutic agents were gathered. 

### 2.4. Patient-Reported Outcome Measures and Neurological Scores 

Patients completed the CIPN toxicity assessment at different time intervals: at baseline before therapy (T0), after anthracycline/cyclophosphamide therapy (T1), after 6 courses of the paclitaxcel treatment (T2), and after 12 courses (T3) of weekly 90 mg/m^2^ paclitaxel (or fewer courses if therapy was aborted early due to side effects) (Figure 1). All neurological assessments were performed by one previously trained researcher. Clinical data were obtained from the patients’ medical records. For patient-reported outcome (PRO) measures, standardized questionnaires were used:-The European Organisation for Research and Treatment of Cancer Quality of Life Questionnaire (EORTC QLQ)-CIPN20 module, a 20-item self-reporting questionnaire containing three subscales to assess sensory, motor, and autonomic CIPN [26]. The total score ranges from 20 to 80; depending on sex and car driving behavior the range can decrease from 18 to 72; higher scores indicate worse CIPN.-The Functional Assessment of Cancer Therapy/Gynecologic Oncology Group-Neurotoxicity (FACT/GOG-NTX) with subscales of physical, social, emotional, and functional wellbeing. The individual scores range from 0 to 28.-The Taxane subscale (TaxS) is a 16-item self-reporting questionnaire focusing on patient-reported neurotoxicity symptoms and concerns. The individual items are scored from 0 to 4 and the sum score ranges from 0 to 64; lower scores indicate worse CIPN [27].-The Trial Outcome Index (TOI) of the FACT-Taxane questionnaire is calculated from physical and functional-wellbeing in addition to the taxane subscale. The score ranges from 0 to 120, with lower scores indicating higher CIPN.-All subscores (physical, social, emotional, functional well-being, and Taxane subscale) together result in the FACT-Taxane Total Score, which ranges from 0 to 172.

Furthermore, neurologic features were evaluated by the Neuropathy Symptom Score (NSS) and Neuropathy Disability Score (NDS) [28]. The NSS includes questions regarding CIPN typical symptoms and scores from 0 to 10. Scores ranging from 0 to 2 were regarded as normal, 3 to 4 as mild symptoms, 5 to 6 as moderate symptoms, and 7 to 10 as severe neuropathy symptoms. The NDS contains typical neurological qualities that can be limited by CIPN. It scores from 0 to 10; scores ranging from 0 to 2 were regarded as normal, 3 to 5 as mild neuropathic signs, 6 to 8 as moderate neuropathic signs, and 9 to 10 as severe neuropathy. 

### 2.5. Assessment of the General Ophthalmological Status

All 12 patients received complete ophthalmological examinations of both eyes, including the determination of the best-corrected distance visual acuity (BCVA) in decimal and converted to logMAR (Logarithm of the Minimum Angle of Resolution), intraocular pressure (ICare tonometer, Icare Finland Oy, Vantaa, Finland), and slit-lamp examinations including a fundoscopy without mydriasis (BM 900 Haag Streit, Möller-Wedel Optical GmbH, Wedel, Germany). Furthermore, patients underwent corneal densitometry with a Scheimpflug system (Pentacam HD, Oculus Optikgeräte GmbH, Wetzlar, Germany) assessment of the autofluorescence of the lens (ClearPath DS-120^®^ Biomicroscope, Freedom Meditech, San Diego, CA, USA) [29,30] and corneal esthesiometry (Cochet-Bonnet, The Luneau Technology Group, Larché, France). Controls received general ophthalmological examinations only. All examinations were performed by the same experienced ophthalmologist throughout the study.

### 2.6. Assessment of Posterior Retinal Segment Imaging and Quantitative Retinal Thickness Analysis 

Macular imaging (fundus imaging) was performed using a laser-based imaging device combining OCT and scanning laser ophthalmoscopy (SPECTRALIS, Heidelberg Engineering GmbH, Heidelberg, Germany) as described before [31,32,33]. Briefly, the volume scan was sized to cover a rectangular area, centered on the fovea, and included consecutive cross-sectional scans. For each examination, care was taken that the size of the volume covered the area for the commonly used early treatment diabetic retinopathy study (ETDRS) grid [34]. After retinal layer segmentation and image registration, dedicated deviation maps were generated by applying an in-house developed visual analysis software to the segmented volume data to visualize and quantify locally resolved thickness differences between selected datasets [32,35,36]. A schematic overview of the generation of deviation maps is shown in Figure 2. The deviation maps were used to compare the retinal thickness of patients at each follow-up time point (T1 to T3) with the baseline time point (T0).

### 2.7. Assessment of Anterior Corneal Segment Imaging and Quantitative Corneal Nerve Analysis 

A combination of the Heidelberg Retina Tomograph 3 (HRT3, Heidelberg Engineering GmbH, Heidelberg, Germany), the Rostock Cornea Module 2.0 (RCM 2.0) [37,38], and EyeGuidance system [39,40] was used for the in vivo confocal laser scanning microscopy (CLSM) assessment of large areas of the corneal subbasal nerve plexus (SNP). Before the measurement, both eyes were anesthetized with topical anesthetic (Proparakain-POS^®^ 0.5% eye drops, Ursapharm, Saarbrücken, Germany) and covered with Vidisic Gel (Bausch & Lomb/Dr. Mann Pharma, Berlin, Germany). This gel was used as an artificial tear and an immersion medium.

During the continuous CLSM image acquisition, the EyeGuidance system presents a fixation target moving on an extending spiral path to the contralateral eye, thus leading to a smoothly guided movement of both eyes. Simultaneously, the focal plane is oscillated by the RCM 2.0 centered on the (manually chosen) initial position at the SNP level with ±20 µm amplitude with a constant speed of 120 µm/s. This imaging procedure generates a volumetric dataset with sufficient SNP images for successful large-area mosaicking. In the mosaicking process, every type of corneal image (epithelium, SNP, and stroma) is used for elastic image registration. Then, a tissue classification algorithm excludes every non-SNP image for the final mosaic. This method has been previously described in detail [41] and is now capable of real-time mosaicking [13]. A schematic overview of the imaging and mosaicking process is shown in Figure 3.

The subsequent image processing and quantitative image analysis were performed by a dedicated algorithm using Mathematica (Version 11.3, Wolfram Research Inc., Champaign, IL, USA), as previously described [42]. The following SNP quantities (Table 1) were calculated: corneal nerve fiber length (CNFL), corneal total nerve fiber density (CTNFD), corneal nerve branching point density (CNBPD), and corneal nerve fiber tortuosity (CNFTo). 

In this longitudinal study, care was taken to evaluate SNP parameters at each time point from the same regions within each patient’s SNP mosaic. For the registration of longitudinal SNP mosaics, static landmarks of the cornea, such as Kobayashi structures [43,44] and nerve entry points, were used as previously described in [45].

### 2.8. Data Management and Statistical Analysis 

Data were analyzed using IBM SPSS 27.0 (Armonk, NY, USA). Significance was defined as *p* < 0.05. Normality was considered when *p* ≥ 0.05 in the Kolmogorov–Smirnov test. All continuous data were expressed as the mean ± SD or as the median (range) for nonparametric samples. Independent sample t-tests (Mann–Whitney-U test for nonparametric samples) were used to demonstrate the differences between patients and the control group. Friedman’s test (for nonparametric samples) was used to evaluate the differences in QLQ and neuropathic scores between patients at baseline and during therapy (follow-up). Categorical variables were reported in frequencies. Fisher’s exact test or Pearson’s Chi-square tests were used to evaluate the association between categorical variables. 

An Analysis of Variance (ANOVA) with repeated measures was used to analyze the longitudinal treatment effects on morphological SNP parameters.

## 3. Results and Discussion

### 3.1. Study Cohort Clinical Characteristics

The clinical characteristics of patients and controls are summarized in Table 2. There was no significant difference in age, body mass index (BMI), and pre-existing conditions such as diabetes, alcohol, or nicotine consumption.

All patients underwent equal therapy strategies with occasional individual differences. The chemotherapy was applied neoadjuvantly, except for two patients that received treatment adjuvantly. The cumulative dose of the individual cytostatic drugs was determined. Chemotherapy was administered in the neoadjuvant setting in 11 patients and as adjuvant treatment in one patient. The tumor and treatment characteristics are summarized in Table 3. In total, eight of the twelve patients were estrogen receptor-negative, and two patients were human epidermal growth factor receptor (HER)2-positive.

### 3.2. Patient-Reported Outcome Measures and Neurological Scores

All patients and controls underwent detailed neurologic and ophthalmologic assessments. The neuropathy findings are reported in Table 4. There was no significant difference between patients and controls in NSS and NDS scores at baseline. Regarding NSS, all patients had no symptoms; one of the ten controls showed mild neurologic symptoms using NSS. Assessment with NDS revealed no neurologic pathology in 12 of the 12 patients and 10 of the 10 controls. Of the various quality of life questionnaires (QLQ), only the FACT-Taxane median scores were significantly lower in patients compared with controls (*p* = 0.009) due to their lower point values in emotional and functional wellbeing (Table 5). 

Over time, all patient scores changed during chemotherapy treatment (Figure 4). From T0 to T3, there was a continuous and significant increase in PRO using the QLQ EORTC-CIPN20 (*p* ≤ 0.001). Accordingly, there was a significant reduction in the Taxane subscale, TOI, and FACT-Taxane total scores (*p* ≤ 0.001, *p* = 0.006, *p* = 0.009, and *p* = 0.046, respectively). The largest differences in neuropathy scores were measured between time T1 and T3 for EORTC-CIPN20 (*p* = 0.006) and the Taxane subscale (*p* = 0.012). The NDS and NSS scores also increased significantly over the study period, with the most marked increase between T1 and T3 (*p* ≤ 0.001 in each case) (Table 6 and Table 7). Frequently reported symptoms and qualities with significant changes during treatment were numbness in the fingertips and toes (*p* = 0.001), general discomfort in feet (*p* ≤ 0.001), pain in fingertips (*p* = 0.019), weakened vibration sense in feet (*p* = 0.015), and weakened or absent patellar and ankle reflexes (*p* = 0.002 and *p* ≤ 0.001). 

Although all assessment methods used showed significant differences in QoL or neurological characteristics before and after treatment, the absolute differences in the various scores are small. For example, the mean NDS at time T3 was 4.0, indicating only mild neuropathic signs, and the proportion of patients with moderate neuropathic signs was 27.3%. On the other hand, the mean NSS at time T3 of 6.6 represents moderate to severe neuropathy with 81.8% of the patients having moderate to severe symptoms. As CIPN is usually symptomatic and the NDS does not reflect symptoms but neurological deficits, the NSS score reflecting symptoms was higher. A strength of the current study is the inclusion of clinical measures (NSS and NDS) and PROs using Taxane-specific QLQ which allows a more comprehensive assessment of CIPN [46]. 

### 3.3. General Ophthalmological Examinations 

In all cases, differences regarding BCVA, intraocular pressure, and morphological changes between the right and left eye were not significant. For subsequent evaluation, the data obtained from the patients’ left eyes were used. The median BCVA was 1 (range 0.8–1) logMAR (T0). The median intraocular pressure (IOP) was 13 (range 10–24) mmHg, and none of the patients were diagnosed with glaucoma. The morphological changes in individuals due to elevated blood sugar levels such as diabetic corneal alterations, neovascularization of the iris (rubeosis iridis), and presenile cataracts were excluded based on a slit lamp examination. Sporadically, only dry eye, conjunctival swelling, papilloma, and benign nevus could be detected. Indirect ophthalmoscopy did not reveal any signs of diabetic retinopathy, i.e., intraretinal hemorrhages, “cotton wool” spots, vascular abnormalities including microaneurysm, and leakages were not found in any eye. The corneal density measured by the dedicated Scheimpflug imaging technique was almost the same in all participants. The median of the autofluorescence ratio of the lens tended to be 0.16 (range 0.1–0.25) at baseline. There were no significant differences in any ophthalmological examination mentioned above over time (T0 to T3) (see Section 2.5).

### 3.4. Thickness of Intraretinal Layers as Investigated by OCT 

OCT measurements were performed on nine patients. The OCT images and measurements of the retinal layers within the macular area were analyzed with dedicated software for group comparisons. Before therapy, all nine patients had no CIPN symptoms and showed mild (NDS) or moderate symptoms (NSS) after therapy (*p* ≤ 0.001). From T0 to T3 the EORTC-QLQ-CIPN20 score increased significantly (*p* ≤ 0.001), while the FACT total score and Taxane subscale (*p* = 0.046; *p* ≤ 0.001) decreased significantly. In OCT, the total retinal and retinal layers including the retinal nerve fiber layer (RNFL), ganglion cell layer (GCL), inner nuclear layer (INL), and inner plexiform layer (IPL) thickness increased significantly (paired T-Test, *p* < 0.05) over time. This was particularly evident in the areas of the parafoveal and perifoveal rings of the ETDRS in comparisons between T0 versus T2 and T0 versus T3 of the total retinal and RNFL thickness. The results of the comparison of retinal thickness for all investigated time points are illustrated in Figure 5 and Figure 6. 

Due to the high metabolic activity of the retina and the high vascularity of the choroid, the interior eye represents an important target of toxicity of anticancer drugs and in many neurological disorders, ocular manifestations precede neurological symptoms [22]. Regarding taxanes such as paclitaxel, ocular side effects include, i.a., optic neuropathy and taxane-induced bilateral cystoid macular edema (CME) [22]. Clinically, taxane-induced maculopathy presents with symptoms including blurred vision and metamorphopsia, an unremarkable anterior segment, and bilateral macular edema with no evidence of vitritis [22]. Until today there have been 16 case reports. Thus, it is not a common side effect and patients demonstrate complete recovery after discontinuation of the drug [22]. The underlying mechanisms are unclear, and several hypotheses have been put forward by investigators. However, unlike other cases of CME, leaking capillaries do not appear to be the source of taxane-induced edema [22]. Among these hypotheses, dysfunctions of Müller cells, which are responsible for maintaining the osmotic gradient in the neurosensory retina, have been proposed. At the same time, other investigators assumed a disruption of microtubular structures in the retinal pigment epithelium (RPE) cells impairing the efficient fluid absorption across the RPE [47,48,49,50]. The present study could demonstrate increasing thickness in retinal layers over time. In a prospective nonrandomized study, the central macular thickness was analyzed by OCT at baseline and at four cycles or at 3 months in visually asymptomatic cancer patients (breast cancer (n = 22), esophageal cancer (n = 2), and ovarian cancer (n = 1)) during taxane-based chemotherapy [51]. All analyzed patients showed no macular abnormalities at baseline, however, after completing four cycles or 12 weeks of chemotherapy, numerous measurements showed an increased retinal thickness [51]. Although the present study included breast cancer patients only and a longer monitoring time interval ranging from baseline to cessation of the therapy, the same trend of increasing retinal thickness as described by Chelala et al. [51] could be detected. The question arose whether these thickenings represent precursors to taxane-induced CME and whether retinal thickenings occur much more frequently than previously assumed during the cancer therapy regime, while the true nature of this phenomenon still remains unclear and would need further investigation. The study shows that monitoring protocols for regular ocular examinations of even asymptomatic cancer patients cannot be overemphasized. Future studies with a larger cohort of patients are highly needed to give more insights into the mechanisms of the thickenings in the retina in general and taxane-induced macular edema, in particular, during cancer therapy. 

### 3.5. Subbasal Nerve Plexus Characteristics

In eight patients, SNP mosaics were successfully recorded for statistical analysis at all four time points. These eight patients were the same as in the OCT analysis, minus one patient who had previously undergone LASIK surgery and was thus excluded from the CLSM. The assessment of the PROs of QoL and the neurological examinations of the eight patients showed the same results as reported above. Figure 7 shows an example of a patient’s SN before (T0), during (T1 and T2), and after completion of therapy (T3). To the best of our knowledge, this is the first study analyzing identical regions within the SNP mosaics within all patients during a period of several months, as outlined in Figure 7 in red. This prevents bias and coincidence in manual image selection using conventional CLSM image selection protocols.

With the herein-used EyeGuidance system, altogether, up to 5 to 10 min per participant were needed for the corneal SNP scanning, increasing the patient’s compliance to participate in regularly recurring monitoring sessions. The average size of the SNP mosaics of all patients was 3.00 mm^2^ ± 0.86 mm^2^, while the average size of identical regions was 1.35 mm^2^ ± 0.77 mm^2^. The mosaics exceeded the size of a standard HRT + CLSM image of 0.16 mm^2^ by about 19 times, while the exact areas exceeded the standard size by about 8 times. Moreover, the conventional CLSM technique depends on the operator’s skills, and imaging the same locations during longitudinal sessions is random or nearly impossible. The EyeGuidance technique overcame these CLSM limitations, opening a new potential for a more objective and realistic real-time in vivo SNP analysis. Figure 8 shows the course of the analyzed nerve parameters (CNFL, CTNFD, CNBPD, and CNFTo) at monitoring sessions T0 to T3. 

In the statistical analysis, the eight patients were not grouped by symptom severity, as this would result in insufficient group sample sizes (two none, four mild, and two moderate). In contrast to the current knowledge, the repeated ANOVA measures applied to all patients without further grouping showed no significant differences for all SNP parameters at any time point of therapy. A recent study by Chinag et al. [19] examined the corneal nerve morphology in patients who had completed neurotoxic paclitaxel or oxaliplatin chemotherapy long after treatment. A sub-analysis was performed in order to investigate the differences between paclitaxel-treated patients with and without neuropathy compared with healthy controls [19]. The corneal nerve parameters were significantly reduced in paclitaxel-treated patients with neuropathy compared with healthy controls and paclitaxel-treated patients without neuropathy. At the same time, paclitaxel-treated patients without neuropathy had values similar to healthy controls [19]. The authors suggested that corneal nerve dysfunction is more pronounced in patients with CIPN, supporting the usefulness of CLSM techniques to monitor nerve function in paclitaxel-treated patients [19]. These findings are also reflected by the present study, as the morphology of the corneal nerves showed no significant differences over time, and the patients did not develop severe CIPN symptoms. 

Interestingly, in a study by Ferdousi et al. [18] using confocal microscopy to identify neuropathy in patients with gastrointestinal cancer before and after platinum-based chemotherapy, CNFL significantly increased after the end of chemotherapy, indicating nerve regeneration. However, this could not be reflected in the present study. Consequently, an increase in sample size is needed for the herein-used method in order to achieve a grouping by symptom severity with sufficient group sample sizes. In future studies, a larger cohort of patients is highly needed to give more insights into the mechanisms of SNP changes during cancer therapy in patients with and without neuropathy.

## 4. Conclusions

The herein-conducted study represents the first longitudinal study combining the oncological examinations of CIPN symptoms with advanced ophthalmological biophotonic imaging techniques. Thus, taxane-based therapy-induced side effects could be directly compared with each other, representing the first longitudinal study with a constant examination of oncological CIPN parameters based on PROs and NDS/NSS, as well as ophthalmological parameters such as corneal nerves and thickness of retinal layers before, during, and after paclitaxel therapy. The OCT measurements revealed retinal thickenings, representing one key breakthrough of the present study. Moreover, this is the first report detecting increasing retinal thickenings in a longitudinal cancer therapy setting, showing that surveillance protocols for regular eye examinations cannot be overestimated even in asymptomatic cancer patients. The second breakthrough of the study was the longitudinal examination of the SNP nerve parameters in mosaics exceeding the size of standard CLSM images. Furthermore, it was possible to identify and analyze identical regions within the SNP within all monitoring sessions. The herein-applied CLSM technology opens the window for future long-term studies in the setting of neuropathy and its effects on peripheral nerves such as corneal nerves. In summary, regular non-invasive in vivo OCT and CLSM monitoring of a patient’s cornea during cancer therapy could complement routine oncological examinations and be helpful in the generation of a more comprehensive clinical picture with benefits for both oncology and ophthalmology. Regarding the future trend of personalized cancer therapy regimens, biophotonic imaging techniques could represent a powerful diagnostic tool for the objective assessment of the severity of adverse events and the outcome of the patient with retinal and corneal structures acting as potential biomarkers. 

## Figures and Tables

**Figure 1 cancers-15-02444-f001:**
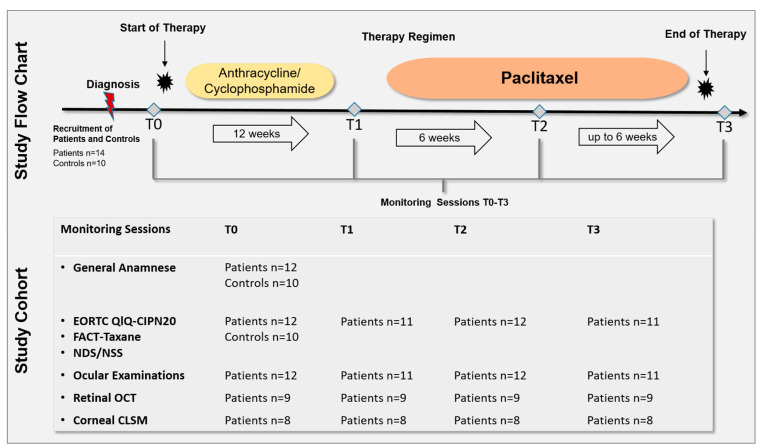
Study flow chart and cohort. A total of 14 patients with newly diagnosed breast cancer and 10 healthy controls were recruited. Of 14 recruited patients, 2 were excluded due to health reasons, therefore, 12 patients and 10 control subjects were assessed. All patients underwent monitoring sessions after diagnosis and before (T0), during (T1 and T2), and after cessation (T3) of the paclitaxel therapy regimen. The monitoring sessions included: the European Organisation for Research and Treatment of Cancer Quality of Life Questionnaire (EORTC QLQ)-CIPN20 and Functional Assessment of Cancer Therapy (FACT)-Taxane questionnaire, Neuropathy Disability Score (NDS), Neuropathy Symptom Score (NSS), ophthalmological status, retinal Optical Coherence Tomography (OCT) of the macula, and corneal Confocal Laser Scanning Microscopy (CLSM) of the corneal subbasal nerve plexus. Because of different therapy durations and individual health conditions, not all patients could be examined at all ophthalmological monitoring sessions. One individual underwent paclitaxel, trastuzumab, and pertuzumab treatment. Controls underwent one monitoring session including general anamneses, EORTC QLQ-CIPN20, FACT-Taxane, NDS, and NSS.

**Figure 2 cancers-15-02444-f002:**
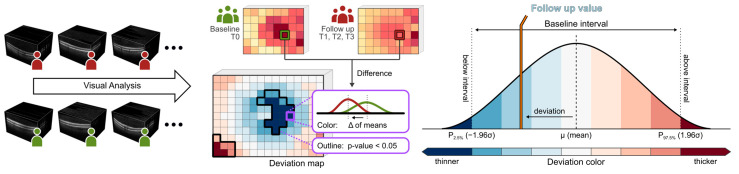
Generation of deviation maps. Point-wise aggregated thickness values were computed per retinal layer and time point from the segmented OCT data of all considered patients. Deviation maps were derived by a point-by-point comparison of the aggregated thickness values of each follow-up time point (T1 to T3) with the baseline time point (T0). The color scale explains the thickness differences of the retinal layers compared via deviation maps. Dark colors indicate values outside the baseline interval, a white color indicates values near the baseline mean, and intermediate colors denote either thinning (blue) or thickening (red). Adjacent points with significant differences (*p* < 0.05) are combined and highlighted by black outlines in the deviation maps.

**Figure 3 cancers-15-02444-f003:**
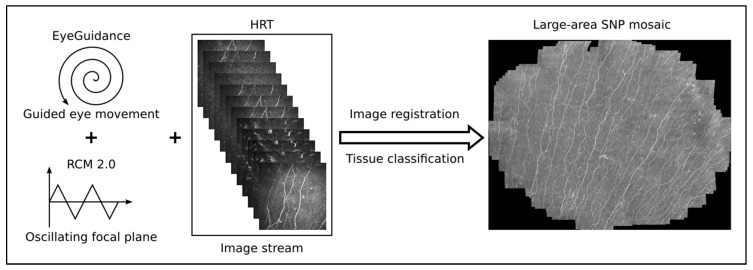
Overview of the imaging and mosaicking process. The combination of EyeGuidance (guided eye movement with spiral pattern), RCM 2.0 (oscillating focal plane), and HRT (continuous image acquisition) captures a volumetric dataset. The final SNP mosaic is created after image registration and tissue classification.

**Figure 4 cancers-15-02444-f004:**
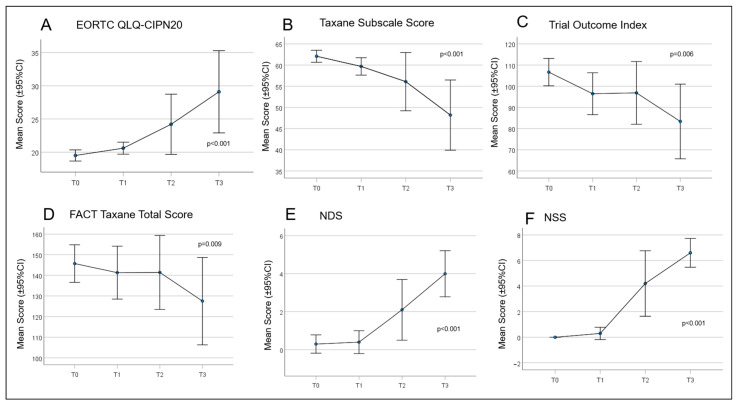
Curves for the patient-reported outcomes (PROs) concerning quality of life (QoL) and paclitaxel-induced neuropathy. (**A**) European Organisation for Research and Treatment of Cancer Quality of Life Questionnaire—Chemotherapy Induced Peripheral Neuropathy (EORTC QLQ-CIPN)20; (**B**) Taxane Subscale Score; (**C**) Trial Outcome Index; (**D**) Functional Assessment of Cancer Therapy (FACT) Taxane Total Score; (**E**) Neuropathy Disability Score (NDS); and (**F**) Neuropathy Symptom Score (NSS).

**Figure 5 cancers-15-02444-f005:**
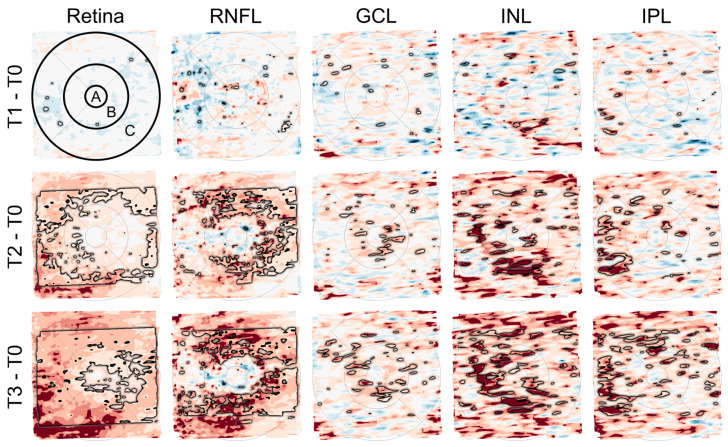
Deviation maps showing differences in the thickness of retinal layers (total retina, retinal nerve fiber layer (RNFL), ganglion cell layer (GCL), inner nuclear layer (INL), and inner plexiform layer (IPL)) between the follow-up time points (T1 to T3) and the baseline time point (T0). The ETDRS grid was used to mark three anatomically distinct areas of the retina (center of the fovea (A) and the parafoveal (B) and perifoveal (C) rings). The rows from top to bottom (T1–T0, T2–T0, and T3–T0) each show a relatively greater number and extent of areas with darker red color, indicating an increase in retinal thickness over time. The black outlines mark areas with significant differences (paired T-Test, *p* < 0.05). This is most evident in the maps for total retinal thickness (first column) and RNFL thickness (second column).

**Figure 6 cancers-15-02444-f006:**
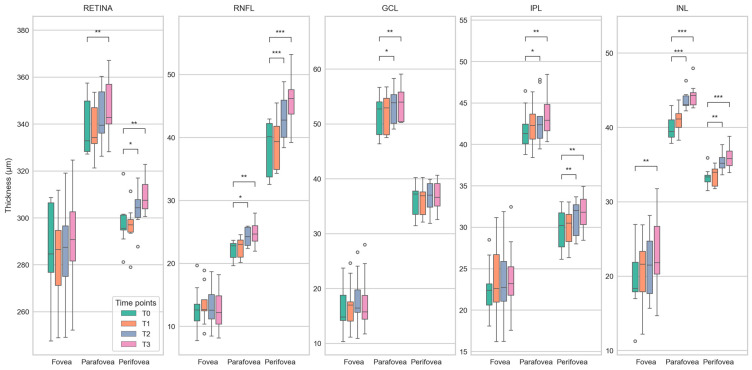
Boxplots showing differences in the thickness of retinal layers (total retina, retinal nerve fiber layer (RNFL), ganglion cell layer (GCL), inner nuclear layer (INL), and inner plexiform layer (IPL)) between baseline (T0) and follow-up (T1 to T3) time points for the center of the fovea and the parafoveal and perifoveal rings of the ETDRS grid. Statistically significant differences (paired T-Test) are marked with *, **, and *** for *p* < 0.05, *p* < 0.01, and *p* < 0.001, respectively. The plots demonstrate a steady increase in the median retinal thickness over time, affecting the measured areas in almost all layers. As with the deviation maps (Figure 5), this is most apparent for total retinal thickness and RNFL thickness.

**Figure 7 cancers-15-02444-f007:**
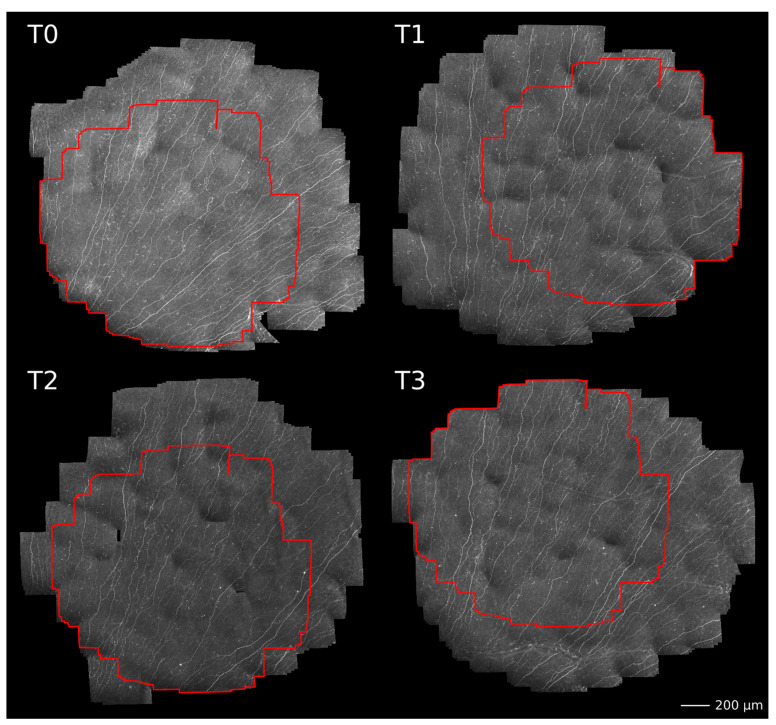
Example of large-area SNP mosaics of one patient acquired before (T0), during (T1 and T2), and after completion of therapy (T3). Please note, the areas outlined in red denote locally identical regions within the SNP that were used for morphological characterization.

**Figure 8 cancers-15-02444-f008:**
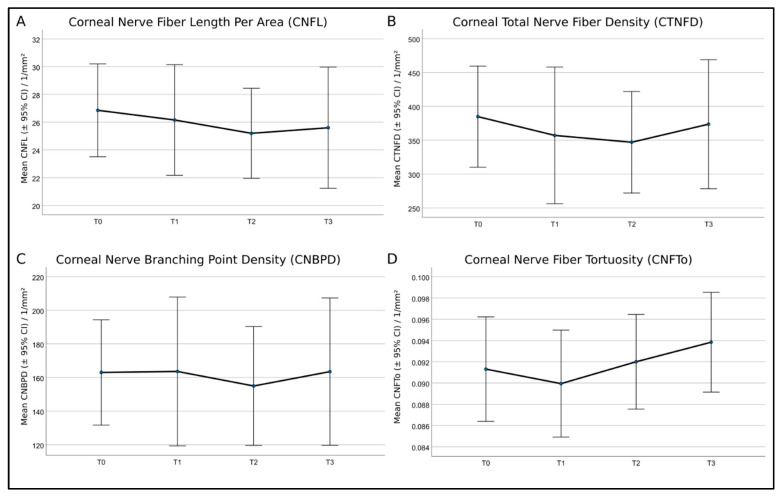
Mean values and 95% confidence intervals (CI) of (**A**)—corneal nerve fiber length per area (CNFL), (**B**)—corneal total nerve fiber density (CTNFD), (**C**)—corneal nerve branching point density (CNBPD), and (**D**)—corneal nerve fiber tortuosity (CNFTo) of the eight patients. Repeated analysis of variance (ANOVA) measures showed no significant differences.

**Table 1 cancers-15-02444-t001:** Definitions of subbasal nerve plexus (SNP) quantities.

SNP Parameter	Abbreviation	Definition	Unit
Corneal nerve fiber length	CNFL	Total length of all nerve fibers per unit area	mm/mm^2^
Corneal total nerve fiber density	CTNFD	Number of nerve fibers per unit area, counting each segment separated by branching points as a single nerve fiber	1/mm^2^
Corneal nerve branching point density	CNBPD	Number of branching points per unit area	1/mm^2^
Corneal nerve fiber tortuosity	CNFTo	(Sum of absolute nerve curvature)/(nerve fiber length)	-

**Table 2 cancers-15-02444-t002:** Clinical characteristics of patients (n = 12) and controls (n = 10).

Characteristics	Patients	Controls	*p*-Value
Sex (n female/n male)	12/0	10/0	n.a.
Median age (range) {years}	49.5 (28–66)	52.5 (27–60)	0.174 ****
Median BMI (range)	24.1 (18.8–40)	22.5 (20.8–26.8)	0.121 ***
Eye disease (n)	4	2	0.646 *
Neurological disease (n)	3	0	0.221 *
Gynecological disease (n)	3	2	1.000 *
Oncological disease (n)	0	0	n.a.
Diabetes (n)	0	0	n.a.
Allergies (n)	7	4	0.670 *
Medication (n)	4	3	1.000 *
Nicotine consumption (n yes/n no more/n no)	4/3/5	0/4/6	0.130 **
Alcohol consumption (n occasionally/n no)	12/0	5/5	0.10 *
Drug consumption (n yes/n no)	0/12	0/10	n.a.

* Fisher test; ** Pearson’s chi-squared test; *** Mann–Whitney U test; **** Student’s *t*-test; BMI: body mass index; n.a.: not applicable.

**Table 3 cancers-15-02444-t003:** Patients’ (n = 12) tumor stage, tumor biology, and treatment characteristics.

Tumor Stage, Grading, and Receptor Status:		n
Stage initial	cT0	1
	cT1	4
	cT2	7
N initial	cN0/pN0	8
	N+ (CNB)	3
	N1 (SLNB)	1
Grading initial	1	1
	2	2
	3	9
ER	positive	4
	negative	8
PR	positive	4
	negative	4
(HER)2	positive	2
	negative	10
**Cytostatic drugs (mean cumulative dose/SD/range) {mg}:**		**n**
Anthracyclin	647.9/62.7/583.2–748.8	11
Cyclophosphamid	4330.4/409.6/3888–4992	11
Paclitaxel	1426.2/418.8/612.0–1996.8	12
Carboplatin	1638.7/846.8/1125–2116	3

T: tumor; N: lymph nodes; CNB: core needle biopsy; SLNB: sentinel lymph node biopsy; ER: estrogen receptor; PR: progesterone receptor; (HER)2: human epidermal growth factor receptor 2; and SD: standard deviation.

**Table 4 cancers-15-02444-t004:** NSS and NDS scores at the time before treatment (T0).

Total Score	Patients (n = 12)	Controls (n = 10)	*p*-Value
NSS (0/1/2/3)	11/1/0/0	9/0/0/1	0.361 **
NDS (0/1/2/3)	10/1/1/0	9/0/1/0	0.645 **

** Pearson chi-squared test; NSS (Neuropathy Symptom Score): 0–2 no symptoms; 3–4 mild symptoms; 5–6 moderate symptoms; and 7–10 severe neuropathy symptoms; NDS (Neuropathy Disability Score): 0–2 no neuropathic signs; 3–5 mild neuropathic signs; 6–8 moderate neuropathic signs; and 9–10 severe neuropathic signs.

**Table 5 cancers-15-02444-t005:** Different CIPN scores at the time before treatment (T0).

CIPN Scores (Median/Range)	Patients (n = 12)	Controls (n = 10)	*p*-Value
EORTC Score	19/18–22	19/19–20	1.000 ***
Physical well being	28/16–28	27/23–28	0.140 ***
Social well being	27/16–28	27.5/25–28	0.203 ***
Emotional well being	17.5/7–20	23/19–24	≤0.001 ***
Functional well being	18.5/8–28	24/22–28	0.009 ***
Taxane Subscale	63/59–64	64/46–64	0.582 ***
Trial Outcome Index	109/87–119	115.5/92–120	0.203 ***
FACT- Taxane total score	149/122–165	164/137–172	0.009 ***

*** Mann–Whitney-U-Test; EORTC: European Organisation for Research and Treatment of Cancer; and FACT: Functional Assessment of Cancer Therapy.

**Table 6 cancers-15-02444-t006:** CIPN scores at different times of chemotherapy (n = 12).

CIPN Scores		T0	T1	T2	T3	*p*-Value *****
EORTC	Mean	19.3	20.4	24.7	28.6	≤0.001
	SD	0.84	1.3	6.2	8.4	
	Range	18–22	19–22	19–40	21–45	
Taxane Subscale	Mean	62.1	59.7	56.1	48.2	≤0.001
	SD	2	2.9	9.6	11.6	
	Range	59–64	55–63	30–63	30–62	
Trial Outcome Index	Mean	106.7	96.5	96.9	83.4	0.006
	SD	9.0	13.8	20.7	24.7	
	Range	87–119	78–116	42–117	40–116	
FACT Taxane Total Score	Mean	145.7	141.3	141.4	127.5	0.046
	SD	12.7	18	25.1	29.6	
	Range	122–161	109–165	75–163	67–162	
NDS total	Mean	0.3	0.4	2.1	4.0	≤0.001
	SD	0.675	0.843	2.234	1.7	
	Range	0–2	0–2	0–6	2–6	
NSS total	Mean	0	0.3	4.2	6.6	≤0.001
	SD	0	0.7	3.6	1.6	
	Range	0	0–2	0–10	4–8	

***** Friedman test; EORTC: European Organisation for Research and Treatment of Cancer; FACT: Functional Assessment of Cancer Therapy; SD: standard deviation; Categories: NDS (Neuropathy Symptom Score): 0–2 no symptoms, 3–5 mild symptoms, 6–8 moderate symptoms, and 9–10 severe neuropathy symptoms; NSS (Neuropathy Disability Score): 0–2 no neuropathic signs, 3–4 mild neuropathic signs, 5–6 moderate neuropathic signs, and 7–10 severe neuropathic signs.

**Table 7 cancers-15-02444-t007:** Friedman test for differences between treatment cycles (*p*-value; n = 10).

Score	T0–T1	T0–T2	T0–T3	T1–T2	T1–T3	T2–T3
EORTC	0.341	0.015	≤0.001	0.141	≤0.001	0.069
Taxane Subscale	0.100	0.69	≤0.001	0.862	0.012	0.019
Trial Outcome Index	0.024	0.194	≤0.001	0.341	0.260	0.038
FACT Total Score	0.488	0.665	0.030	0.26	0.141	0.009
NDS	1.000	0.119	≤0.001	0.119	≤0.001	0.038
NSS	0.665	0.019	≤0.001	0.057	≤0.001	0.141

EORTC: European Organisation for Research and Treatment of Cancer; FACT: Functional Assessment of Cancer Therapy; NDS: Neuropathy Disability Score; and NSS: Neuropathy Symptom Score.

## Data Availability

The data presented in this study are available on request from the corresponding author.

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
