# Peer review of "Taxane-Induced Neuropathy and Its Ocular Effects—A Longitudinal Follow-up Study in Breast Cancer Patients"

_cancers, 2023, doi:10.3390/cancers15092444_

Round 1

Reviewer 1 Report

This paper details a longitudinal study utilising various neurological and ophthalmic assessments to characterise peripheral neuropathic and ocular nerve changes in patients treated with neurotoxic chemotherapy, namely paclitaxel. It is a highly relevant and well-thought out piece as it is novel in incorporating quantitative measures of both corneal nerves and retinal findings obtained using sophisticated image analysis methods. In the current field of oculomics or ocular biomarkers, this article would greatly contribute to our knowledge in both the clinical and research settings, highlighting the awareness of subtle ocular nerve changes in these patients for medical oncologists, neurologists and ophthalmic clinicians. Only a few minor concerns remain:

1. In the introduction (line 44, page 1), the authors may have made a mistake in stating '4.179' rather than '4,179' or '4179'. The authors should clarify and correct this.

2. In line 72, page 2 of the introduction, the authors mentioned that 'no prospective longitudinal studies which monitor cancer patients from baseline to follow-up after cancer therapy exist'. Could the authors briefly or compare their results with and/or comment on findings from previous studies by Riva et al (Riva et al. Corneal and Epidermal Nerve Quantification in Chemotherapy Induced Peripheral Neuropathy. Front Med (Lausanne). 2022 Feb 18;9:832344) which looked at a variety of neurotoxic drugs before and after treatment including paclitaxel? This could be done in the discussion, and the statement in the introduction changed to reflect this.

3. In line 80, examples of side effects especially those relevant to the drug being investigated in this paper should be stated here. e.g. taxane-induced macula edema.

4. In line 353, page 11 of the results section, could the authors clarify whether the 'p<0.05' is referring to the measures of all areas including foveal, parafoveal and perifoveal regions? In line with this comment, Figure 6 should incorporate some indication of which measures or areas should significant changes. Perhaps this could be done in the caption should doing so in the figure itself be too crowded or confusing.

5. In line 396, page 12, the author's name for reference 50 should be referred to before the reference number: '...thickness as described by Chelala et al [50]...'.

Reviewer 2 Report

This is a comprehensive clinical and ophthalmic assessment of the effects of paclitaxel in patients with breast cancer over 24 weeks at 3 time points.

The patients were phenotyped in detail for neurotoxicity with various questionnaires including  (EORTC QLQ)-CIPN20 module; (FACT/GOG-NTX), (TaxS), NSS and NDS.

The patients underwent detailed ophthalmic assessment with OCT and HRTIII with wide mosaic quantification of the sub-basal plexus.

The only limitation is the small number of patients studied 9-12.

NDS is not a measure of symptoms, but neurological deficits which are large fibre weighted and a score of 4 is indicative of mild neuropathy. The NSS was higher as CIPN is usually symptomatic.

What is the explanation for an increase in total retinal and retinal layer 351 (retinal nerve fiber layer (RNFL), ganglion cell layer (GCL), inner nuclear layer (INL), and 352 inner plexiform layer (IPL)) thickness?

The lack of overall change in corneal nerves is not consistent with previous data. Although there does seem to be a reduction initially with an increase, which would be consistent with initial degeneration followed by regeneration as found by Ferdousi et al PLoS One 2015, 10, doi:10.1371. This merits further discussion.This is a comprehensive clinical and ophthalmic assessment of the effects of paclitaxel in patients with breast cancer over 24 weeks at 3 time points.

The patients were phenotyped in detail for neurotoxicity with various questionnaires including  (EORTC QLQ)-CIPN20 module; (FACT/GOG-NTX), (TaxS), NSS and NDS.

The patients underwent detailed ophthalmic assessment with OCT and HRTIII with wide mosaic quantification of the sub-basal plexus.

The only limitation is the small number of patients studied 9-12.

NDS is not a measure of symptoms, but neurological deficits which are large fibre weighted and a score of 4 is indicative of mild neuropathy. The NSS was higher as CIPN is usually symptomatic.

What is the explanation for an increase in total retinal and retinal layer 351 (retinal nerve fiber layer (RNFL), ganglion cell layer (GCL), inner nuclear layer (INL), and 352 inner plexiform layer (IPL)) thickness?

The lack of overall change in corneal nerves is not consistent with previous data. Although there does seem to be a reduction initially with an increase, which would be consistent with initial degeneration followed by regeneration as found by Ferdousi et al PLoS One 2015, 10, doi:10.1371. This merits further discussion.

Reviewer 3 Report

The paper treats a very important topic: chemotherapy induced peripheral neuropathy (CIPN) and neurotoxic effect of BC therapy on eyes. It is a  longitudinal clinical trial that associate oncological examinations with advanced biophotonic imaging techniques. It seems a perfectly useful tecnique to better tailor treatments.

No concerns
